# The causal effect of workplace violence on mental health and work-related outcomes: a cross-sectional study using propensity score matching

Javier A. Flores-Cohaila[1], Brayan Miranda-Chavez[2,3], Cesar Copaja-Corzo [4,5]*

**1** Grupo de Investigación en Healthcare Simulation & Medical Education (HeSIM), Facultad de Ciencias de la Salud, Universidad Científica del Sur, Lima, Perú, **2** Centro de Estudios e Investigación en Educación Médica y Bioética, EDUCAB-UPT, Facultad de Ciencias de la Salud, Universidad Privada de Tacna, Tacna, Perú, **3** Servicio de Geriatría, Hospital Nacional Guillermo Almenara Irigoyen, EsSalud, Lima, Perú, **4** Unidad de Investigación para la Generación y Síntesis de Evidencias en Salud, Universidad San Ignacio de Loyola, Lima, Perú, **5** Servicio de infectologia, Hospital Nacional Edgardo Rebagliati Martins, EsSalud, Lima, Perú

* Csarcopaja@gmail.com

## Abstract

### Introduction

The prevalence of workplace violence in healthcare is 50–60%. While it has been linked to decreased job satisfaction, diminished quality of care, and economic burdens on healthcare systems, there are still major gaps. Previous studies ignored the Latin American perspective. Moreover, they neither offered causal evidence nor measured its impact on psychological outcomes. The objective was to evaluate the impact of workplace violence on psychological and work-related outcomes.

### Methods

A secondary analysis of Peru's 2016 National Healthcare Satisfaction Survey was conducted. This was a large-scale survey that used a stratified two-stage cluster sample design with a sample size of 5098 healthcare workers across all regions of Peru. Propensity score matching and Poisson regression models were used to assess the effect of self-reported workplace violence on outcomes, including depressive symptoms, burnout, sleep problems, work-life balance, and intention to quit.

### Results

Among 4,951 healthcare workers, workplace violence prevalence was 41.91% higher in physicians (47.4%) than nurses (37.8%). WV had a moderate effect on sleep problems (aPR: 2.06, 95% CI: 1.45 to 2.97) and depressive symptoms (aPR: 1.65, 95% CI: 1.47–1.86). It showed small to moderate effects on burnout dimensions and

**Data availability statement:** All relevant data are within the paper and its Supporting Information files.

**Funding:** The Universidad San Ignacio de Loyola finances the article processing charge. Funding Acquisition was by author C.C.C. The funders had no role in study design, data collection and analysis, decision to publish, or preparation of the manuscript.

**Competing interests:** The authors have declared that no competing interests exist.

intention to quit (aPR: 1.26, 95% CI: 1.13–1.41). The impact on work-life balance was small to negligible.

## Conclusions

Workplace violence affects 4 in 10 Peruvian healthcare workers and is associated with adverse psychological and work-related outcomes. These findings highlight the need for improved reporting systems, targeted interventions such as policy development and training programs, and ensure adequate reporting systems.

---

## Introduction

Workplace violence (WV) poses a major challenge for healthcare. Its prevalence stands at 50–60% among healthcare workers (HCWs), with psychological being the most common [1–4]. It affects several life dimensions of HCWs. WV is associated with stress and job dissatisfaction. It also impacts the healthcare system, leading to diminished quality of care [5–9] and even carries an economic burden [10]. Hence, several organizations have prioritized addressing WV in healthcare [11,12].

The field remains unclear despite the global recognition of workplace violence (WV) as a healthcare priority. Our current understanding of WV is marked by a global north perspective, which may not reflect a diverse cultural context [2,11]. Furthermore, most of the research has focused on reporting the prevalence of WV, with little effort being put into studying its impact [8]. Moreover, there is a lack of exploration of causality in studies that report an association with different outcomes, which is understandable due to ethical constraints and a lack of advanced statistical approaches. Lastly, while some authors intended to provide causal evidence, the outcomes of the studies were only on psychological effects [13,14].

Hence, there is an unmet need to understand WV worldwide. At the same time, other studies have closed the gap of WV in neglected regions, such as Africa [15]. However, there is still a lack of research in the Latin American region. Moreover, this region has high rates of WV, mostly affecting HCWs [16]. Among the region's countries, Peru stands out due to major challenges such as its high diversity, lack of HCWs, fragmentation with different funding sources, and multiple health service delivery channels, which led to the burden of HCWs and inequity for the population [17]. These claims make Peru an interesting setting for studying WV and its impact on yet unaddressed outcomes.

Hence, this study evaluates the impact of workplace violence on psychological and work-related outcomes. Moreover, to strengthen the validity of our findings and provide richer insights, we employ advanced causal inference techniques, such as propensity score matching (PSM) [18]. This technique enables researchers to mimic randomization through adjustment of cofounders and is of most importance when the data is obtained through observational studies [19]. Moreover, as exposure to WV in a randomized controlled trial is unethical, the PSM is the most adequate method to assess its impact on eligible outcomes.

This study's findings will help fill the gap on WV's impact on HCWs' psychological outcomes, such as depressive symptoms, burnout, and sleep problems, and work-related outcomes, such as intention to quit and work-life balance. Hence, they will help to inform evidence-based policies and interventions, contributing to the global effort to address this pervasive issue in healthcare settings.

## Materials and methods

### Study design and data sources

This secondary analysis is based on the 2016 National Healthcare Satisfaction Survey (ENSUSALUD, for its abbreviation in Spanish), the last ENSUSALUD survey conducted so far. It was organized by the National Health Superintendence and the Minister of Health from Peru and conducted by the National Institute of Statistics and Informatics from April to July 2016 [20]. This survey was chosen because it is the largest one at present. Moreover, due to its stratified two-stage cluster sample design and funding, it remains superior to primary data collection.

Peru is a diverse country with roughly 34 million inhabitants across 24 geopolitical regions, also called departments. Its healthcare system is fragmented across different sectors, such as the Minister of Health, Social Insurance, Army/Police, and Private Institutions. It is structured across levels of complexity based on resolution capacity, with I-1 being the lowest and III-E the highest.

For this large-scale survey, a stratified two-stage cluster sample design was followed. The primary sampling unit was health facilities (n = 183), and the secondary sampling unit was physicians and nurses (n = 5098) across all regions of Peru. The dataset is available online (https://portal.susalud.gob.pe/blog/base-de-datos-2016/).

The author, J.F.C., accessed the website on May 3, 2024, downloaded the anonymous dataset, and imported it into the RStudio program. The dataset was cleaned and filtered using the dplyr package in RStudio.

### Variables and procedures

The outcome variables were: 1) Depressive symptoms, measured with the Patient Health Questionnaire (PHQ)-2 [21], which consists of two items. Response options include "never," "some days," "more than half of days," and "nearly every day." The total score is calculated by summing items, ranging from 2 to 8. Scores higher or equal to 3 are considered "depressive symptoms." 2) Burnout and its dimensions, measured with the Maslach Burnout Inventory [22]. It is a 22-item questionnaire comprised of three domains: Professional fulfillment, emotional exhaustion, and depersonalization. Each domain is interpreted independently. We considered a cut-off score for "signs of burnout" of below 34 for professional fulfillment, above 9 for depersonalization, and above 26 for emotional exhaustion; 3) Sleep problems, measured with the Jenkins Sleep Scale, a 4-item questionnaire [23]. Items are rated 1–6 based on frequency in the last month: 1 = never, 6 = almost every night. The cut-off score for considering a high frequency of sleep disturbances is 11 (higher); 4) Worklife balance, which was measured by asking the question, "Does your workload give you enough time for personal and family life?" It was scored on a 5-Likert scale, with 1 = totally disagree and 5 = totally agree. For this study, we considered scores 4 and 5 as "good work-life balance" and 1–3 as "no work-life balance."; and 5) Intention to quit, measured with the question, "Do you think or have plans to leave work in this establishment?". The options were yes, no, or do not know. For this study, we eliminated "does not know." All of these conversions were done based on previous literature [24–27].

The independent variable in this study was Workplace Violence (WV), measured through self-reported experiences within the past 12 months, with the question, "In the last 12 months, have you been a victim of … in your workplace?". The four types of WV were physical aggression, threats, insult, or sexual harassment. Response options included "yes," "no," or left blank. For blank responses, we recorded them as "no" for the analysis. In conducting regression analysis, we created a variable called "any type of workplace violence." Participants were classified as "yes" for this variable if they reported any of the four types of workplace violence mentioned earlier; in the absence of any type of WV, they were classified as "no."

The following covariates were included: age (years), gender (male or female), job (nurse or physician), postgraduate training - referring to pursuing a master's or doctoral degree (yes or no), level of care (primary, secondary or tertiary), marital status (single or married), family support (yes or no), work stability based on contract (yes or no), years of practice (junior or senior, based on a cut-off of ten years), working hours (based on a cut-off of 40 hours per week), income (categorized as less than 3,000 new soles [$ 800], between 3,001 and 7,500 soles [$800–2,000] and over 7,500 soles), and region of residence (Lima or other regions)

## Sample size estimation

While previous studies have been conducted [28], there were none to estimate the sample size required for our study. Hence, we conducted a post-hoc power size calculation with the G*Power 3.1 software [29]. We assumed a minimum detectable effect size of 1.5, with a marginal exposure probability (WV) of 0.6 [4], a statistical power of 80%, and a false positive rate of 0.05. Since we planned to use propensity score matching (PSM), we assumed equal proportions for the intervention and control groups (50% each). Additionally, the marginal probability of the outcome was set at 0.8, resulting in a minimum required sample size of 3,979 participants.

## Data analysis

We followed a three-step process for data analyses: estimation of the prevalence of WV and its types, propensity score matching, and regression analysis to evaluate the effect of WV on psychological outcomes. Analyses were conducted in RStudio Version 4.2.2 (Viena, Austria, RStudio Corporation), employing the MatchIt packages [30]. The code and the dataset are available as supporting information.

First, the prevalence and 95% Confidence Interval (95% CI) for WV and its types were estimated. For this, we considered the cases of WV as the numerator and the total population as the denominator. Moreover, the 95% CI was calculated using the binomial test. This was performed for overall WV, its types for the whole population and subgroups, and the outcomes.

Second, propensity score matching (PSM) using the nearest method with a 1:1 matching method was employed, with violence as the outcome and the covariates, described above, as independent variables [18]. The closeness of the propensity scores of treatment and control units matched them. Units without a match were excluded. Chi-square and T-student tests were employed to identify differences between treatment and control units.

Third, we employed Poisson regression models adjusting for all covariates. This was done for all healthcare workers and for subgroups of physicians/doctors and nurses. The models were performed on pre-matching and post-matching datasets. All models were assessed for colinearity with the Variance inflation factor, and all were below 5. We employed forest plots to represent the effect sizes.

## Ethical considerations

This study used anonymous data from the openly accessible ENSUSALUD 2016 survey. Participant confidentiality was ensured, and only those who gave informed consent were included. The research adhered to the ethical principles of the Declaration of Helsinki throughout the process.

## Results

As shown in Fig 1, a total of 5098 HCWs were surveyed in ENSUSALUD 2016. We did not include 147, including 91 physicians and 56 nurses, mostly due to missing data (2.88%) (Fig 1).

## Patient characteristics and workplace violence

Table 1 portrays key characteristics and the prevalence of WV and its subtypes. The mean age was 45.08 (SD [Standard deviation]: 11.27), ranging between 23 and 76. Most were females (62.88%), nurses (57.08%), and working in

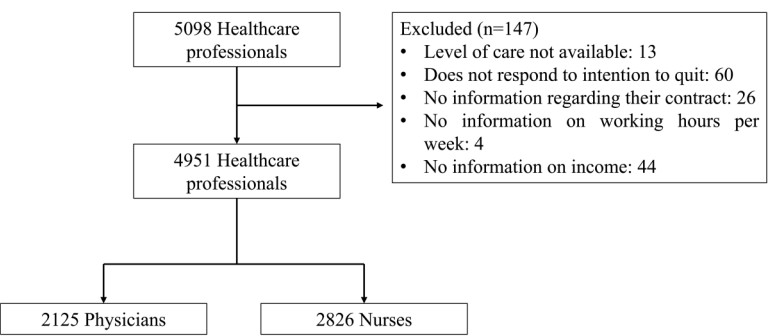

**Fig 1. Participant selection flowchart.**

the secondary level of care (56.17%). Less than half of HCWs had postgraduate training (39.57%). Most were married (33.61%) and had work stability (69.84%). Income was around 3,001 and 7,500 new soles, which is between 1,000–2,000 dollars in the majority (57.24%). Most HCWs were from regions outside Lima (77.60%).

Self-reported WV prevalence was 41.91 (95% CI: 40.53 to 43.28), higher in physicians than nurses (47.4 versus 37.8). Insults were the most common form of WV violence. The prevalence of self-reported WV was highest among those with an income of over 7,501 new soles (51.09%), followed by physicians, males, those working in primary care, and those who worked over 40 hours. The lowest prevalence was seen in those with lower incomes (36.67%) and nurses (37.8%), females (39.35%), and those with work stability (40.18%). Regarding subtypes of violence, higher income had the highest prevalence of self-reported violence in all subtypes, except physical violence, where tertiary care took the lead.

As shown in Table 2, physicians had a higher prevalence of all psychological and work outcomes than nurses. Four out of 10 HCWs reported having no work-life balance, while three out of 10 reported wanting to quit their current jobs. Almost one in four HCWs experienced depressive symptoms and emotional exhaustion.

## Workplace violence effect on psychological and work outcomes

The results of the chi-squared and T-student test for socio-demographic characteristics separated by pre and post-PSM are shown in Table 3. As shown in the PSM, there were no differences between categories.

As shown in Fig 2, adjusted multivariate Poisson regression models were estimated for each outcome of interest based on the pre and post-PSM. WV was moderately associated with depressive symptoms in all models (PSM: aPR: 1.65, 95% CI: 1.47 to 1.86), as well as the intention to quit (PSM: aPR: 1.26, 95% CI: 1.13 to 1.41). In the case of professional fulfillment, WV had a small to no effect on physicians (aPR: 1.00, 95% CI: 0.64 to 1.66), while it was moderate in nurses (aPR: 1.92, 95% CI: 1.38 to 2.47). WV had a small effect on depersonalization and emotional exhaustion. However, it moderately affected sleep problems for physicians (aPR: 2.06, 95% CI: 1.45 to 2.97) and nurses (aPR: 2.18, 95% CI: 1.48 to 3.28). Lastly, the effect on work-life balance was small in physicians (aPR: 1.18, 95% CI: 1.04 to 1.34) to null in nurses (aPR: 1.16, 95% CI: 1.00 to 1.34).

## Discussion

### Summary of findings

Here, we conducted a secondary analysis of a large dataset that comprises 4951 HCWs (2125 doctors and 2826 nurses) from Peru to identify the effect of workplace violence on psychological outcomes, such as depressive symptoms, burn-out, sleep problems and work-related ones, such as work-life balance and intention to quit, employing propensity score matching. Our major findings are as follows: 1) WV prevalence stands at 41.91 (95% CI: 40.53 to 43.28), being higher in

**Table 1. Characteristics of participants and prevalence of workplace violence and types per 100 Healthcare workers.**

| Category | N (%) | WV (95% CI) | Insults (95% CI) | Threats (95% CI) | Physical (95% CI) | Sexual (95% CI) |
|---|---|---|---|---|---|---|
| **Gender** | | | | | | |
| Female | 3113 (62.88) | 39.35 (37.63 to 41.09) | 31.93 (30.29 to 33.60) | 20.17 (18.77 to 21.63) | 6.49 (5.65 to 7.41) | 1.32 (0.95 to 1.78) |
| Male | 1838 (37.12) | 46.25 (43.95 to 48.56) | 37.86 (35.64 to 40.12) | 28.89 (26.83 to 31.02) | 8.81 (7.56 to 10.20) | 2.39 (1.74 to 3.20) |
| **Job** | | | | | | |
| Physician | 2125 (42.92) | 47.4 (45.3 to 49.6) | 38.54 (36.46 to 40.64) | 29.69 (27.75 to 31.68) | 8.42 (7.27 to 9.68) | 2.40 (1.79 to 3.14) |
| Nurse | 2826 (57.08) | 37.8 (36.0 to 39.6) | 30.82 (29.12 to 32.56) | 18.68 (17.26 to 20.17) | 6.54 (5.66 to 7.52) | 1.20 (0.83 to 1.68) |
| **Posgraduate training** | | | | | | |
| No | 2992 (60.43) | 41.41 (39.64 to 43.19) | 34.09 (32.39 to 35.82) | 22.66 (21.17 to 24.20) | 7.05 (6.16 to 8.03) | 1.64 (1.21 to 2.16) |
| Yes | 1959 (39.57) | 42.67 (40.47 to 44.90) | 34.20 (32.09 to 36.35) | 24.55 (22.66 to 26.52) | 7.81 (6.66 to 9.09) | 1.84 (1.29 to 2.54) |
| **Level of care** | | | | | | |
| Primary | 575 (11.61) | 45.22 (42.00 to 49.39) | 35.82 (31.90 to 39.90) | 27.83 (24.20 to 31.68) | 6.61 (4.72 to 8.96) | 2.26 (1.21 to 3.84) |
| Secondary | 2781 (56.17) | 40.70 (38.87 to 42.56) | 34.52 (32.75 to 36.32) | 21.79 (20.27 to 23.37) | 6.29 (5.42 to 7.26) | 1.55 (1.12 to 2.08) |
| Tertiary | 1595 (32.22) | 42.82 (40.38 to 45.29) | 32.85 (30.55 to 35.22) | 24.64 (22.54 to 26.83) | 9.47 (8.07 to 11.01) | 1.81 (1.22 to 2.60) |
| **Marital status** | | | | | | |
| Married | 3287 (66.39) | 42.04 (40.35 to 43.75) | 34.07 (32.45 to 35.72) | 23.43 (21.99 to 24.91) | 7.18 (6.32 to 8.12) | 1.64 (1.24 to 2.14) |
| Single | 1664 (33.61) | 41.65 (39.26 to 44.06) | 34.25 (31.97 to 36.59) | 23.38 (21.36 to 25.49) | 7.69 (6.46 to 9.08) | 1.86 (1.27 to 2.63) |
| **Family support** | | | | | | |
| Yes | 4286 (86.57) | 41.48 (40.00 to 42.98) | 33.71 (32.30 to 35.15) | 22.49 (21.25 to 23.77) | 7.21 (6.45 to 8.03) | 1.56 (1.21 to 1.98) |
| No | 665 (13.43) | 44.66 (40.84 to 48.53) | 36.84 (33.16 to 40.64) | 29.32 (25.89 to 32.94) | 8.27 (6.29 to 10.62) | 2.71 (1.61 to 4.24) |
| **Work stability based on contract** | | | | | | |
| Yes | 3458 (69.84) | 42.65 (40.99 to 44.32) | 34.24 (32.66 to 35.85) | 24.00 (22.59 to 25.46) | 8.10 (7.21 to 9.06) | 1.85 (1.43 to 2.36) |
| No | 1493 (30.16) | 40.18 (37.69 to 42.73) | 33.89 (31.49 to 36.36) | 22.04 (19.96 to 24.23) | 5.63 (4.51 to 6.92) | 1.41 (0.87 to 2.14) |
| **Years of practice** | | | | | | |
| Junior (≤10 years) | 2891 (58.39) | 41.89 (40.09 to 43.71) | 34.35 (32.62 to 36.11) | 23.97 (22.42 to 25.57) | 6.68 (5.79 to 7.65) | 1.73 (1.29 to 2.27) |
| Senior (>10 years) | 2060 (41.61) | 41.94 (39.80 to 44.11) | 33.83 (31.79 to 35.92) | 22.62 (20.83 to 24.49) | 8.30 (7.15 to 9.58) | 1.70 (1.19 to 2.36) |
| **Working hours in a week** | | | | | | |
| Less or equal than 40 hours | 2718 (54.90) | 39.88 (38.03 to 41.75) | 32.82 (31.05 to 34.52) | 20.20 (18.70 to 21.76) | 6.73 (5.82 to 7.74) | 1.51 (1.08 to 2.04) |
| Over 40 hours | 2233 (45.10) | 44.38 (42.30 to 46.47) | 35.74 (33.75 to 37.76) | 27.32 (25.48 to 29.22) | 8.11 (7.01 to 9.32) | 1.97 (1.44 to 2.64) |
| **Income** | | | | | | |
| Less than 3,000 new soles | 1475 (29.79) | 36.67 (34.21 to 39.19) | 30.10 (27.77 to 32.51) | 18.78 (16.82 to 20.87) | 5.89 (4.75 to 7.22) | 0.81 (0.42 to 1.42) |
| From 3,001–7,500 new soles | 2834 (57.24) | 42.55 (40.73 to 44.39) | 34.62 (32.86 to 36.39) | 23.64 (22.09 to 25.25) | 7.94 (6.97 to 8.99) | 1.87 (1.40 to 2.44) |
| Over 7,501 new soles | 642 (12.97) | 51.09 (47.15 to 55.02) | 41.28 (37.44 to 45.19) | 33.02 (29.39 to 36.81) | 8.09 (6.11 to 10.49) | 3.12 (1.91 to 4.77) |
| **Region** | | | | | | |
| Lima | 1109 (23.39) | 43.91 (40.97 to 46.89) | 34.27 (31.47 to 37.14) | 24.89 (22.37 to 27.54) | 9.74 (8.06 to 11.64) | 1.80 (1.10 to 2.77) |
| Other regions | 3842 (77.60) | 41.33 (39.77 to 42.91) | 34.09 (32.59 to 35.62) | 22.98 (21.66 to 24.35) | 6.66 (5.89 to 7.49) | 1.69 (1.31 to 2.15) |

WV - Workplace Violence, CI - Confidence Interval

**Table 2.** Prevalence and 95% Confidence Intervals of depressive symptoms, burnout, sleep problems, work-life balance, and intention to quit in Peruvian physicians and nurses.

| Outcomes | All (n = 4951) | Physicians (n = 2125) | Nurses (n = 2826) |
|---|---|---|---|
| Depressive symptoms | 26.82 (25.59 to 28.06) | 26.96 (25.08 to 28.85) | 26.72 (25.08 to 28.35) |
| No work-life balance | 39.06 (37.70 to 40.42) | 47.11 (44.98 to 49.23) | 33.02 (31.28 to 34.79) |
| Burnout: Low professional fulfillment | 2.66 (2.22 to 3.11) | 2.96 (2.24 to 3.69) | 2.44 (1.19 to 3.01) |
| Burnout: Depersonalization | 37.10 (35.76 to 38.45) | 44.94 (42.83 to 47.05) | 31.21 (29.50 to 32.92) |
| Burnout: Emotional exhaustion | 26.82 (25.59 to 28.06) | 33.98 (31.96 to 35.99) | 21.44 (19.93 to 22.96) |
| Sleep problems | 5.59 (4.95 to 6.24) | 6.59 (5.53 to 7.64) | 4.85 (4.06 to 5.64) |
| Intention to quit | 28.19 (26.94 to 29.45) | 33.65 (31.64 to 35.65) | 24.09 (22.52 to 25.67) |

**Table 3.** Comparison of the covariates after propensity score matching.

| Category | Before PSM | | | After PSM | | |
|---|---|---|---|---|---|---|
| | Non-WV (N = 2876) | WV (N = 2075) | x2 (p-value) | Non-WV (n = 2075) | WV (n = 2075) | x2 (p-value) |
| Age | 45.17 (11.49) | 44.97 (10.95) | 0.531 | 45.19 (11.42) | 44.97 (10.95) | 0.520 |
| Gender: Male | 988 (34.4) | 850 (41.0) | <0.001 | 856 (41.3) | 850 (41.0) | 0.875 |
| Job: Nurse | 1759 (61.2) | 1067 (51.4) | <0.001 | 1041 (50.2) | 1067 (51.4) | 0.438 |
| Posgraduate training: Yes | 1123 (39.0) | 836 (40.3) | 0.394 | 827 (39.9) | 836 (40.3) | 0.800 |
| Level of care | | | 0.091 | | | 0.328 |
| Primary | 315 (11.0) | 260 (12.5) | | 229 (11.0) | 260 (12.5) | |
| Secondary | 1649 (57.3) | 1132 (54.6) | | 1150 (55.4) | 1132 (54.6) | |
| Tertiary | 912 (31.7) | 683 (32.9) | | 696 (33.5) | 683 (32.9) | |
| Marital status: Single | 971 (33.8) | 693 (33.4) | 0.812 | 672 (32.4) | 693 (33.4) | 0.509 |
| Family support: Yes | 2508 (87.2) | 1778 (85.7) | 0.133 | 1796 (86.6) | 1778 (85.7) | 0.445 |
| Work stability based on contract: Yes | 1983 (68.9) | 1475 (71.1) | 0.113 | 1470 (70.8) | 1475 (71.1) | 0.891 |
| Years of practice: Junior | 1680 (58.4) | 1211 (58.4) | 0.994 | 1210 (58.3) | 1211 (58.4) | 1.000 |
| Working hours in a week: Over 40 hours | 1242 (43.2) | 991 (47.8) | 0.002 | 972 (46.8) | 991 (47.8) | 0.576 |
| Income | | | <0.001 | | | 0.615 |
| Less than 3,000 new soles | 934 (32.5) | 541 (26.1) | | 534 (25.7) | 541 (26.1) | |
| From 3,001–7,500 new soles | 1628 (56.6) | 1206 (58.1) | | 1233 (59.4) | 1206 (58.1) | |
| Over 7,501 new soles | 314 (10.9) | 328 (15.8) | | 308 (14.8) | 328 (15.8) | |
| Region: Other regions | 2254 (78.4) | 1588 (76.5) | 0.134 | 1599 (77.1) | 1588 (76.5) | 0.713 |

physicians than nurses (47.4 versus 37.8); 2) Insults were the most common form of WV, followed by threats, physical and sexual violence; 3) WV had a moderate effect on the development of sleep problems and low personal fulfillment; however, the later was only seen in nurses; 4) WV had a small to moderate effect on the development of depressive symptoms, depersonalization, and emotional exhaustion; and 5) WV had a small effect on the perception of no work-life balance and in the intention to quit.

## Limitations and strengths

Some limitations need to be considered when interpreting our results. First, we relied on a secondary dataset that was not collected to address our research questions. Hence, although we adjusted for several cofounders, some may still be

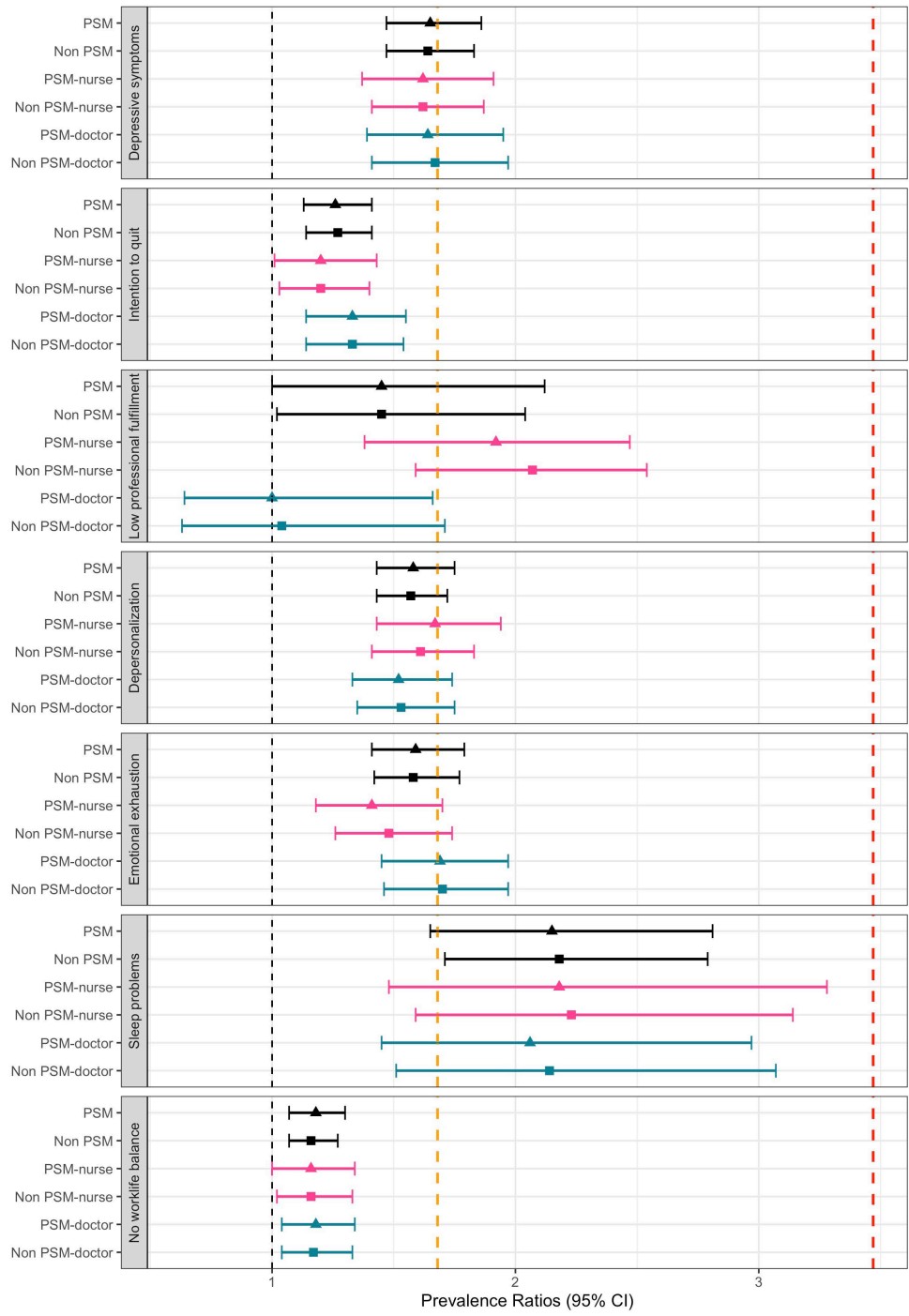

**Fig 2. Effect sizes on the Poisson multivariate regression with and without propensity score matching in healthcare workers, doctors/physicians, and nurses.** PSM - Propensity score matching. Black dashed line: Line of no effect at 1.00; Orange dashed line: Line of small effects at 1.68; Red dashed line: Line of moderate effect at 3.47.

unconsidered. Second, WV was self-reported and may introduce bias due to social desirability or fear of consequences [31]. Hence, the prevalence of workplace violence may be higher than we reported. Lastly, while we employ PSM, the cross-sectional nature of the study design limits the ability to make strong causal inferences.

Despite these limitations, this study has some strengths, such as the large sample size, the adherence to reporting guidelines, and the employment of causal inference techniques.

## Comparison with prior work

In our study, 4 out of 10 HCWs experienced WV in the past 12 months, slightly below those reported in systematic and umbrella reviews. Notably, the prevalence of sexual violence in our study (1.72%) was much lower than the global rate of 12.4% [1,2,4]. Additionally, contrary to previous research, our findings indicated a higher prevalence of all types of WV among physicians than nurses [2]. Two key explanations emerge. First, previous studies suggest that WV may be perceived as "part of the job" [32,33], potentially leading to non-reporting. Second, underreporting may also result from fear of repercussions or social desirability bias [8]. In a context with a high prevalence of chauvinism, it is particularly unlikely that the true prevalence of sexual violence is below the global average [34]. Thus, the true estimate of WV may be higher than reported.

WV had a moderate to small effect on sleep problems, depressive symptoms, depersonalization, and emotional exhaustion. This is in accordance with a wealth body of literature [13,35,36]. WV is a significant stressor, affecting even those who witness it. This is particularly concerning in healthcare, where mental health disorder prevalence exceeds that of the general population [37]. Therefore, WV may intensify these conditions. Moreover, the problems described here may represent only the surface, as WV has been linked to post-traumatic stress disorder and substance abuse [38–40]. Symptoms like depressive signs and burnout may reflect underlying disorders.

There was a small to negligible effect on the intention to quit. This differs from one large study in Switzerland on 1,441 HCWs that found that those who experienced WV had a moderate to large effect on the intention to quit [41]. Two major explanations arise for this difference. First, that study did not account for cofounders nor employ causal inference techniques. Second, major differences between Switzerland and Peru range from culture to priorities in life to salary. In the Latin American context, work-related laws and job security are far from what developed countries may consider standards. This may explain why Peruvian HCW may not intend to quit even in deplorable conditions. Hence, it may not only be due to statistical concerns but also depends on the context, which requires further research.

## Implications

This is the first large study to evaluate the effect of WV on several psychological and work outcomes in the South American region. Therefore, it has several implications. While the prevalence of WV was high, the comparison with worldwide estimates suggests an underreporting of some types of violence, such as sexual and physical. Hence, further studies are needed to uncover the true prevalence. Moreover, it requires the study of potential reasons why there may be an underreporting. Potential avenues comprise qualitative studies to address how WV is perceived, as previous research has shown that HCW's experiences it as"part of the job."[32]

According to our study, WV's impact is detrimental and needs to be considered by policymakers. While this is not a recent problem, previous researchers have stated its importance. However, little to no efforts have been made in Peru. While our findings provide a portrayal of the effects of WV, there is a need for longitudinal studies to assess long-term consequences such as attrition or migration. There is a need to explore other psychological outcomes, such as post-traumatic stress disorder, as it has been delineated in the literature, and it is feasible that the psychological outcomes reported here may be a sign of other mental health problems as a product of WV.

Lastly, considering the impact of WV on the healthcare system, actions must be taken urgently. A recent review of interventions to reduce workplace violence shows that most fall under two categories: prevention and de-escalation [42]. While prevention interventions focused on identifying workplace violence through risk assessment skills, the de-escalation interventions comprised training on communication skills to de-escalate patients. Notably, that review found no studies from the Latin America region. Hence, new interventions, as mentioned before, need to be adopted or developed. Lastly, there is still no light on our findings and the research landscape in WV.

## Conclusions

Our findings suggest that workplace violence is moderately associated with depression, burnout, and insomnia while slightly associated with no work-life balance and leave intention. This study highlights the impact of workplace violence in the healthcare system and represents a call to implement better reporting systems, implement interventions, and conduct further research in this realm.

## Supporting information

**S1 Appendix. Dataset.**
(DOCX)

**S2 Appendix. RStudio Script.**
(DOCX)

**S3 Appendix. Supplementary table 1.**
(XLSX)

## Author contributions

**Conceptualization:** Javier A. Flores-Cohaila, Brayan Miranda-Chavez.

**Data curation:** Javier A. Flores-Cohaila, Brayan Miranda-Chavez, Cesar Copaja-Corzo.

**Formal analysis:** Javier A. Flores-Cohaila.

**Funding acquisition:** Cesar Copaja-Corzo.

**Investigation:** Brayan Miranda-Chavez.

**Methodology:** Javier A. Flores-Cohaila, Cesar Copaja-Corzo.

**Writing – original draft:** Javier A. Flores-Cohaila, Brayan Miranda-Chavez, Cesar Copaja-Corzo.

**Writing – review & editing:** Javier A. Flores-Cohaila, Brayan Miranda-Chavez, Cesar Copaja-Corzo.

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
