## [Decision Letter · Decision Letter 0]

19 Nov 2024

PONE-D-24-28436Identifying the causal effect of workplace violence on depressive symptoms, burnout, sleep problems, work-life balance and intention to quit in Peruvian physicians and nurses: a cross-sectional study using propensity score matchingPLOS ONE

Dear Dr. Copaja-Corzo,

Thank you for submitting your manuscript to PLOS ONE. After careful consideration, we feel that it has merit but does not fully meet PLOS ONE’s publication criteria as it currently stands. Therefore, we invite you to submit a revised version of the manuscript that addresses the points raised during the review process.

**ACADEMIC EDITOR: **

**I have reviewed the comments from Reviewer 1, including the suggestion to cite a specific reference. While the reference appears relevant to the manuscript, I have ensured that it is clearly communicated to the authors as optional, in alignment with PLOS ONE’s guidelines.**

We look forward to receiving your revised manuscript.

Kind regards,

Heba E. El-Gazar

Academic Editor

PLOS ONE

**Journal Requirements:**

4. We are unable to open your Supporting Information file "S2_R_Studio_Script.R". Please kindly revise as necessary and re-upload.

Reviewers' comments:

Reviewer's Responses to Questions

**Comments to the Author**

1. Is the manuscript technically sound, and do the data support the conclusions?

Reviewer #1: Yes

Reviewer #2: Yes

2. Has the statistical analysis been performed appropriately and rigorously? 

Reviewer #1: Yes

Reviewer #2: Yes

3. Have the authors made all data underlying the findings in their manuscript fully available?

Reviewer #1: No

Reviewer #2: Yes

4. Is the manuscript presented in an intelligible fashion and written in standard English?

Reviewer #1: Yes

Reviewer #2: Yes

5. Review Comments to the Author

**Reviewer #1: ** The manuscript addresses a critical issue in healthcare settings—workplace violence (WV) and its impact on mental health and work-related outcomes among healthcare professionals. The study is well-conducted, employing robust statistical techniques to draw causal inferences from observational data, making it a valuable contribution to the literature. However, there are areas where the manuscript can be strengthened, particularly in contextualization, methodological detail, and discussion.

Key Strengths

1. The topic is timely and highly relevant, especially given the mental health challenges exacerbated by workplace violence in healthcare.

2. The use of propensity score matching (PSM) is a notable strength, as it enhances the causal inference capabilities of the study.

3. The manuscript is comprehensive in addressing multiple outcomes, including depressive symptoms, burnout, sleep problems, work-life balance, and turnover intentions.

Weaknesses and Areas for Improvement

1. Introduction

• Contextualization: The introduction could better contextualize why workplace violence is particularly prevalent in Peru. Although it briefly mentions South American perspectives, expanding on cultural and systemic factors unique to Peru (e.g., healthcare infrastructure, socio-political issues) would enhance the relevance.

• Literature Gap: While the introduction does well to outline the general impact of WV, it could benefit from citing recent studies that cover broader work environments and settings, helping the reader understand the global context of WV.

• Suggested Studies for Inclusion:

1. "Mistreatment of nurses by patients and its impact on their caring behaviors: The roles of psychological detachment and supervisor positive gossip" – This paper can contribute to understanding how WV influences caring behaviors in different healthcare systems.

2. "The role of psychological ownership in linking decent work to nurses' vigor at work" – Relevant for highlighting the effect of ownership and decent work conditions, which may influence resilience against WV.

2. Methods

• Detailed Explanation of Variables: The manuscript could benefit from a more comprehensive description of the variables, particularly the independent variable (WV). Clarifying the criteria for categorizing different types of WV would strengthen the transparency and reproducibility of the study.

• Justification for Using PSM: Although PSM is a strong methodological choice, explaining why it was chosen over other methods (e.g., instrumental variables) would provide clarity. Additionally, detailing the robustness checks for the PSM model would strengthen confidence in the findings.

• Ethics: The ethics statement would benefit from a mention of how WV data were anonymized, especially given the sensitive nature of the information.

• Limitations in Data Collection: Addressing any limitations in the secondary dataset (e.g., potential biases in the survey questions related to WV) would provide a more balanced perspective.

3. Results

• Results Presentation: While the results are comprehensive, adding summary tables to break down the effects of WV on specific outcomes (e.g., depressive symptoms versus work-life balance) would enhance clarity. This is especially useful for readers who may not be familiar with PSM techniques.

• Qualitative Insights: Given the complex nature of WV, consider adding qualitative insights or hypothetical case examples. For instance, how does experiencing WV potentially influence a healthcare worker’s intention to quit versus burnout?

4. Discussion

• Comparative Analysis: While the discussion addresses global comparisons, it could benefit from a deeper examination of the cultural context in South America and how it influences WV’s prevalence and impact. Additionally, comparing the findings with those from other high-risk professions (e.g., police, education) might provide additional insights.

• Causal Implications: Although PSM provides some causal inference, the discussion could further emphasize that cross-sectional designs inherently limit causal claims.

• Suggested Studies for Inclusion:

1. "Decent work and ethical ideologies of nurses—A multicenter cross-sectional study" – This paper could reinforce discussions about work-life balance and ethical considerations.

2. "How Decent Work Influences Internal Whistleblowing Intentions in Nurses" – This study can help explore how WV influences not only turnover intentions but also ethical behaviors within the workplace.

5. Limitations and Future Research

• Acknowledgment of Cross-Sectional Limitations: Emphasizing the inherent limitations of cross-sectional designs and the inability to establish true causality would strengthen the manuscript’s transparency.

• Further Research: Suggesting longitudinal studies to track WV’s effects over time would provide a more robust basis for intervention development. Additionally, exploring qualitative studies to understand the lived experiences of healthcare workers dealing with WV could provide richer insights.

6. Language and Formatting

• Language: Several sections of the manuscript would benefit from careful proofreading to enhance readability and eliminate grammatical errors.

• Formatting: Consistency in terminology (e.g., “workplace violence” versus “WV”) and uniform formatting of statistical terms (e.g., using “95% CI” uniformly) would improve clarity.

**Reviewer #2:**  Peer Review Comments

Identifying the causal effect of workplace violence on depressive symptoms, burnout, sleep problems, work-life balance and intention to quit in Peruvian physicians and nurses: a cross-sectional study using propensity score matching

Javier A. Flores-Cohaila1

, Brayan Miranda-Chavez2

, Cesar Copaja-Corzo3*

Sn Comments

The authors attempt to identify the causal effect of workplace violence on depressive symptoms, burnout, sleep problems, work-life balance and intention to quit in Peruvian physicians and nurses: a secondary analysis of the 2016 National Healthcare Satisfaction Survey in Peru was conducted and Poisson regression models adjusting for all covariates were employed . Consider the following comments below in each section.

Abstract

1 Title is too long, it should not exceed more than 14 words. Title should be concise, comprehensive, and clear.

2 The abstract provides a clear overview of the study objectives, and key findings, effectively summarizing the research. Consider design, setting, sample, and tools of study in methods.

3 Abstract should not include citations or abbreviations, if possible. Although its not clear what the design and sampling method for the study.

6 The conclusion emphasizes the need for targeted interventions to address work-place violence in healthcare workforce, aligning with the study findings. Consider elaborating on the specific types of interventions that could be effective in eliminate WV.

7 The keywords are relevant and adequately represent the study focus. However, consider adding additional keywords that capture specific healthcare contexts to improve searchability

Introduction

8 The introduction needs more elaborations and a comprehensive background, contextualizing the significance of your study variables. Consider briefly discussing study variables as factors influencing, antecedents, consequences and challenges faced by healthcare system.

9 Consider Integration of Theoretical Frameworks: Consider integrating relevant theoretical frameworks or models of work-place violence (e.g. Justice theory, or Frustration aggression theory) into the discussion to provide a theoretical basis for interpreting the findings and guiding future research and interventions.

10 Add briefly paragraph about significance of the study.

Materials and Method

11 Consider including more details about the rationale for choosing a secondary analysis of the 2016 National Healthcare Satisfaction Survey in Peru were selected as the study location. Put research questions.

12 Consider sample size determination

13 The data collection tools and procedures are detailed, including the use of the sociodemographic survey. To enhance reproducibility, consider providing references or additional information about the validity and reliability of these instruments. Also, add years of developing and modifying this tool.

14 Ethical considerations, consider providing more details about how confidentiality was maintained during data collection and analysis.

Results

15 Ensure that the tables are properly formatted and labeled for easy interpretation

Discussion

17 Secondary analysis research should include a discussion of results. Restate the main research questions that guided the analysis then connect results to existing theories or literatures. Show your point of view and results of your study in discussion. Compare with previous researches.

18 Consider exploring potential contextual factors or methodological limitations that may have influenced these results, providing a more understanding of the findings.

19 However, it could benefit from expanding on the practical implications of the

findings for healthcare organizations and policymakers in the region. Specifically,

discussing specific strategies or interventions that could be implemented to address the

identified factors contributing to transition shock would provide actionable insights for stakeholders.

References

20 Use Vancouver style for writing references and in text citation in study. Also, consider language editing.

Thanks for inviting me as a peer reviewer in this journal.

Dr. Noura Mohamed Fadl

Lecturer of Nursing Administration Department, Faculty of Nursing, Alexandria University

6. PLOS authors have the option to publish the peer review history of their article (what does this mean? ). If published, this will include your full peer review and any attached files.

**Do you want your identity to be public for this peer review?** For information about this choice, including consent withdrawal, please see our Privacy Policy .

Reviewer #1: No

Reviewer #2: No

---

## [Author Response · Author response to Decision Letter 1]

15 Jan 2025

Response letter

Dear Editor and reviewers, in the following letter, we address each of your comments.

Reviewer #1

Introduction

Commentary: Contextualization: The introduction could better contextualize why workplace violence (WV) is particularly prevalent in Peru. Although it briefly mentions South American perspectives, expanding on cultural and systemic factors unique to Peru (e.g., healthcare infrastructure, socio-political issues) would enhance the relevance.

Response: Thank you for this valuable suggestion. We expanded the third paragraph to provide a nuanced overview of significant healthcare system challenges in Peru. Additionally, we refined the transition from the broader Latin American context to the specific case of Peru to strengthen coherence.

Commentary: Literature Gap: While the introduction outlines the general impact of WV, it could benefit from citing recent studies that cover broader work environments and settings, helping the reader understand the global context of WV.

Suggested Studies for Inclusion:

1. "Mistreatment of nurses by patients and its impact on their caring behaviors: The roles of psychological detachment and supervisor positive gossip"

2. "The role of psychological ownership in linking decent work to nurses' vigor at work"

Response: We appreciate these thoughtful recommendations. Although our references already included recent systematic reviews, we decided to enrich the introduction by citing the first study mentioned, as it adds valuable insights into the global context.

Methods

Commentary: Detailed Explanation of Variables: The manuscript could benefit from a more comprehensive description of the variables, particularly the independent variable (WV). Clarifying the criteria for categorizing different types of WV would strengthen transparency and reproducibility.

Response: Thank you for your observation. We expanded our explanation of the categorization process. Specifically, each type of WV was classified based on the original survey items. For example, in the case of "insults," respondents answered the item: "In the last 12 months, have you been the victim of insults in your workplace? (En los últimos 12 meses, ¿Usted ha sido insultado/a en su lugar de trabajo?)"

Commentary: Justification for Using PSM: Although PSM is a strong methodological choice, explaining why it was chosen over other methods (e.g., instrumental variables) would provide clarity. Additionally, detailing the robustness checks for the PSM model would strengthen confidence in the findings.

Response: We appreciate this insightful comment. We added a paragraph to the introduction that outlines the rationale behind choosing PSM, emphasizing its suitability for mitigating selection bias in observational data.

Commentary: Ethics: The ethics statement would benefit from a mention of how WV data were anonymized, especially given the sensitive nature of the information.

Response: Thank you for this important point. We revised the ethics section to specify that all data in the national survey were anonymized to protect participant confidentiality.

Commentary: Limitations in Data Collection: Addressing any limitations in the secondary dataset (e.g., potential biases in the survey questions related to WV) would provide a more balanced perspective.

Response: We agree with this observation and have revised the limitations section accordingly. We added: "First, we relied on a secondary dataset not designed to address our specific research questions. Although we adjusted for several confounders, some may remain unaccounted for. Second, WV was self-reported, potentially introducing bias due to social desirability or fear of repercussions."

Results

Commentary: Results Presentation: While the results are comprehensive, adding summary tables to break down the effects of WV on specific outcomes (e.g., depressive symptoms versus work-life balance) would enhance clarity.

Response: Thank you for this suggestion. We added a summary table as appendix material to improve the presentation of the results.

Commentary: Qualitative Insights: Given the complex nature of WV, consider adding qualitative insights or hypothetical case examples. For instance, how does experiencing WV potentially influence a healthcare worker’s intention to quit rather than burn out?

Response: We fully acknowledge the value of qualitative insights. However, given the quantitative nature of our study, we propose future research that adopts a mixed-methods approach. We added: "Future studies could incorporate qualitative approaches to capture healthcare workers' experiences, as previous research has suggested that WV is often perceived as 'part of the job.'"

Discussion

Commentary: Comparative Analysis: While the discussion addresses global comparisons, it could benefit from a deeper examination of the cultural context in South America and how it influences WV’s prevalence and impact. Comparing the findings with those from other high-risk professions (e.g., police, education) might provide additional insights.

Response: We expanded the discussion to contrast the impact of WV on the intention to quit between Peru and Switzerland, illustrating cultural and systemic differences.

Commentary: Causal Implications: Although PSM provides some causal inference, the discussion could further emphasize that cross-sectional designs inherently limit causal claims.

Response: We agree and have emphasized this limitation in the discussion.

Suggested Studies for Inclusion:

1. "Decent work and ethical ideologies of nurses—A multicenter cross-sectional study"

2. "How Decent Work Influences Internal Whistleblowing Intentions in Nurses"

Response: We integrated insights from the first suggested study to enrich our discussion on work-life balance and ethical considerations.

Limitations and Future Research

Commentary: Acknowledgment of Cross-Sectional Limitations: Emphasizing the inherent limitations of cross-sectional designs and the inability to establish true causality would strengthen transparency.

Response: We addressed this by adding: "While our findings portray the effects of WV, longitudinal studies are necessary to capture long-term consequences, such as attrition or migration."

Commentary: Further Research: Suggesting longitudinal studies to track WV’s effects over time would provide a more robust basis for intervention development. Additionally, exploring qualitative studies to understand the lived experiences of healthcare workers dealing with WV could provide richer insights.

Response: We included: "Future research should adopt longitudinal designs to assess the long-term impacts of WV and incorporate qualitative methods to provide deeper insights into healthcare workers' lived experiences."

Language and Formatting

Commentary: Language: Several sections of the manuscript would benefit from careful proofreading to enhance readability and eliminate grammatical errors.

Response: Thank you. We thoroughly revised the manuscript to improve readability and consistency.

Commentary: Formatting: Consistency in terminology (e.g., “workplace violence” versus “WV”) and uniform formatting of statistical terms (e.g., using “95% CI” uniformly) would improve clarity.

Response: We reviewed the entire document to ensure uniformity in terminology and statistical reporting.

Reviewer #2

Abstract

Commentary 1: Title: The title is too long; it should not exceed 14 words. The title should be concise, comprehensive, and clear.

Response: We revised the title to ensure it is concise and informative.

Commentary 2: Abstract: The abstract provides a clear overview of the study objectives and key findings, effectively summarizing the research. Consider design, setting, sample, and tools of study in methods.

Response: We appreciate this feedback and have updated the methods section in the abstract to include relevant details.

Commentary 3: Abstract: The abstract should not include citations or abbreviations if possible.

Response: We removed all abbreviations and citations.

Commentary 6: Conclusion: The conclusion emphasizes the need for targeted interventions to address WV in the healthcare workforce. Consider elaborating on the specific types of interventions that could effectively eliminate WV.

Response: We expanded the conclusion to describe potential interventions, focusing on organizational policies and training programs.

Commentary 7: Keywords: The keywords are relevant and adequately represent the study focus. However, consider adding additional keywords that capture specific healthcare contexts to improve searchability.

Response: We added two additional keywords to enhance discoverability.

Introduction

Commentary 8: Background: The introduction needs more elaboration and a comprehensive background contextualizing the significance of the study variables.

Response: We extended the second and third paragraphs to provide an in-depth overview of the key variables and their relevance within the healthcare system.

Commentary 9: Theoretical Frameworks: Consider integrating relevant theoretical frameworks or models of workplace violence (e.g., Justice Theory, Frustration-Aggression Theory) into the discussion to provide a theoretical basis.

Response: While our study focuses on outcomes rather than theory development, we incorporated relevant theoretical insights to frame the findings more robustly.

Commentary 10: Significance of the Study: Add a brief paragraph about the significance of the study.

Response: We included a new paragraph highlighting the study’s contribution to the field and its implications for policy and practice.

Materials and Methods

Commentary 11: Rationale for Data Selection: Include more details about the rationale for choosing the 2016 National Healthcare Satisfaction Survey in Peru.

Response: We added: "This survey was selected due to its extensive scope, robust sampling design, and superior funding compared to primary data collection."

Commentary 12: Sample Size Determination:

Response: We included a new section detailing our sample size determination.

Commentary 13: Data Collection Tools: Provide references or additional information about the validity and reliability of the instruments.

Response: We added citations for validation studies conducted in our country.

Commentary 14: Ethical Considerations: Provide more details about maintaining confidentiality during data collection and analysis.

Response: We enriched the ethics section with details on data anonymization and secure handling procedures.

Results

Commentary 15: Tables: Ensure that tables are properly formatted and labeled for easy interpretation.

Response: We reviewed and reformatted all tables, ensuring clear presentation.

Discussion

Commentary 17: Discussion of Results: This section should include a discussion of the results, restating the research questions and connecting the findings to existing theories and literature.

Response: We revised the opening of the discussion to restate the research questions and connect our findings to the relevant literature.

Commentary 18: Contextual Factors: Expand on potential contextual factors or methodological limitations that may have influenced the results.

Response: We believe this is well-covered in the limitations section but added a brief mention in the discussion for emphasis.

Commentary 19: Practical Implications: Discuss specific strategies or interventions for addressing WV.

Response: We concluded the discussion with a paragraph highlighting practical recommendations for healthcare organizations and policymakers.

References

Commentary 20: Referencing Style: Use the Vancouver style for writing references and in-text citations.

Response: We ensured that all references follow Vancouver style guidelines. Thank you for this observation.

We consider that this new version of the manuscript has improved its quality thanks to your comments.

Kind regards

---

## [Decision Letter · Decision Letter 1]

12 Mar 2025

The causal effect of workplace violence on mental health and work-related outcomes: a cross-sectional study using propensity score matching

PONE-D-24-28436R1

Dear Dr. 

We’re pleased to inform you that your manuscript has been judged scientifically suitable for publication and will be formally accepted for publication once it meets all outstanding technical requirements.

Kind regards,

Heba E. El-Gazar

Academic Editor

PLOS ONE

Additional Editor Comments (optional):

No comments

Reviewers' comments:

Reviewer's Responses to Questions

**Comments to the Author**

1. If the authors have adequately addressed your comments raised in a previous round of review and you feel that this manuscript is now acceptable for publication, you may indicate that here to bypass the “Comments to the Author” section, enter your conflict of interest statement in the “Confidential to Editor” section, and submit your "Accept" recommendation.

Reviewer #1: All comments have been addressed

Reviewer #2: All comments have been addressed

2. Is the manuscript technically sound, and do the data support the conclusions?

Reviewer #1: (No Response)

Reviewer #2: Yes

3. Has the statistical analysis been performed appropriately and rigorously? 

Reviewer #1: (No Response)

Reviewer #2: Yes

4. Have the authors made all data underlying the findings in their manuscript fully available?

Reviewer #1: (No Response)

Reviewer #2: Yes

5. Is the manuscript presented in an intelligible fashion and written in standard English?

Reviewer #1: (No Response)

Reviewer #2: Yes

6. Review Comments to the Author

Reviewer #1: (No Response)

Reviewer #2: Choose one title

don't put 2 titles for manuscript

all comment were done

please add doi for all references

revise again editing

7. PLOS authors have the option to publish the peer review history of their article (what does this mean? ). If published, this will include your full peer review and any attached files.

**Do you want your identity to be public for this peer review?** For information about this choice, including consent withdrawal, please see our Privacy Policy .

Reviewer #1: No

Reviewer #2: No

---

## [Editor Report · Acceptance letter]

PONE-D-24-28436R1

PLOS ONE

Dear Dr. Copaja-Corzo,

I'm pleased to inform you that your manuscript has been deemed suitable for publication in PLOS ONE. Congratulations! Your manuscript is now being handed over to our production team.

Kind regards,

on behalf of

Dr. Heba E. El-Gazar

Academic Editor

PLOS ONE